# Towards continuous learning for glioma segmentation with elastic weight consolidation

**Karin van Garderen** [1,2]                    k.vangarderen@erasmusmc.nl

**Sebastian van der Voort** [1,3]               s.vandervoort@erasmusmc.nl

**Fatih Incekara**[1]                           f.incekara@erasmusmc.nl

**Marion Smits**[1]                             marion.smits@erasmusmc.nl

**Stefan Klein**[1,3]                           s.klein@erasmusmc.nl

[1] *Erasmus MC, Dept. of Radiology and Nuclear Medicine, Rotterdam, the Netherlands*

[2] *Medical Delta, Delft, the Netherlands, www.medicaldelta.nl*

[3] *Erasmus MC, Dept. of Medical Informatics*

## Abstract

When finetuning a convolutional neural network (CNN) on data from a new domain, catastrophic forgetting will reduce performance on the original training data. Elastic Weight Consolidation (EWC) is a recent technique to prevent this, which we evaluated while training and re-training a CNN to segment glioma on two different datasets. The network was trained on the public BraTS dataset and finetuned on an in-house dataset with non-enhancing low-grade glioma. EWC was found to decrease catastrophic forgetting in this case, but was also found to restrict adaptation to the new domain.

**Keywords:** convolutional neural network, glioma, segmentation, continuous learning, magnetic resonance imaging

## 1. Introduction

Automatic segmentation of glioma from MR imaging is a relevant problem that can be solved by convolutional neural networks (CNNs). These models require large annotated datasets to train and they can be sensitive to domain shift, for example when used in a new center with a different scanner (Ghafoorian et al., 2017). Ideally, the model would be re-trained with annotated data from the new domain, improving the model and adapting to a new domain continuously. However, a neural network will suffer from catastrophic forgetting of the information from previous datasets if they are not also included in this re-training (McCloskey and Cohen, 1989).

This leads to the question: can we implement continuous learning, so that a CNN can benefit from additional data without having access to the original dataset? This is a relevant question in the medical field where sharing data is often difficult due to privacy concerns and the amount of data a single institution can gather about a specific condition is limited by its prevalence.

In this study we evaluate the use of Elastic Weight Consolidation (EWC) (Kirkpatrick et al., 2017) to overcome catastrophic forgetting when re-training a network on new data. EWC penalizes large changes in the weights through an additional weighted term in the loss function, which is specifically computed for each parameter based on its importance for the

original dataset. It has been evaluated previously for sequential learning of structure and lesion segmentation on brain MRI (Baweja et al., 2018). In this study, we evaluate it in a domain transfer setting with two similar datasets and a single task: glioma segmentation.

## 2. Methods

The methods are evaluated on two datasets: the 2018 BraTS Challenge training set containing 285 subjects with low- and high-grade glioma (source domain) and an in-house dataset containing 98 subjects with non-enhancing low-grade glioma (target domain). The images contain four MR sequences: pre- and post-contrast T1-weighted, T2-weighted, and T2-weighted FLAIR. All volumes were co-registered, skull-stripped and resampled to 1 mm$^3$ voxel size. The voxel intenstities were normalized to zero mean and unit standard deviation.

Although the datasets are prepared in a similar way, the inclusion criteria are very different. The BraTS data contains mostly tumors of a higher grade and with contrast enhancement, while the target domain contains only non-enhancing low-grade glioma. As a result, the mean tumor volume is three times smaller and they are often more difficult to distinguish (see Fig. 1). Also, the target dataset may contain images with a lower quality and the segmentations were performed by a single observer while the BraTS dataset was segmented by one to four raters.

A 3D UNet CNN with unpadded convolutions was trained and evaluated in different training schemes. The network has a field of view of $88^3$ voxels. For training, random patches were extracted of $108^3$ voxels with a ground-truth segmentation of $20^3$ voxels. One network was trained with both domains simultaneously for 100 epochs. Another network was trained first for 50 epochs on the source domain and then for 50 epochs on the target domain. This re-training of the network was performed with and without EWC penalty on the loss function, with an EWC weight factor of 10 and 100. The model was evaluated on a held-out test set containing 20% of each of the datasets. The experiments were performed with two different random splits.

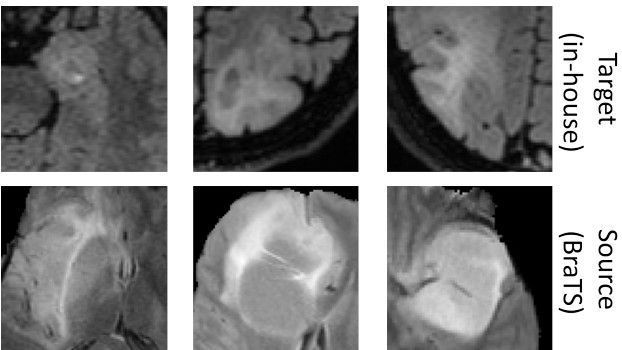

Figure 1: Examples from the target dataset (top) and compared to low-grade tumors from the BraTS dataset (bottom). Images are cropped in the axial slice around the tumor on the T2W-FLAIR sequence.

## 3. Results

The performance results are presented in Table 1. The baseline performance for the target domain was lower than the performance on BraTS data in general. The EWC penalty improved performance on the source domain after continuing the training on the target domain. However, it also restricted the adaptation capacity to the target domain. Training with two datasets together did not lead to better results, and especially for the target domain it was more effective to train specifically on each of the datasets.

Table 1: Performance results in mean Dice percentage for two random splits of the dataset. Source is the BraTS dataset and Target is the in-house dataset of non-enhancing glioma pre-processed with skull-stripping. The + operator indicates a re-training procedure of the model pre-trained on source only.

| Training | Split 1 | | Split 2 | |
|---|---|---|---|---|
| | Source | Target | Source | Target |
| Target only | 64 | 71 | 60 | 74 |
| Source only | 86 | 58 | 79 | 58 |
| + Target | 68 | 73 | 63 | 75 |
| + Target EWC 10 | 78 | 72 | 62 | 74 |
| + Target EWC 100 | 73 | 69 | 72 | 72 |
| Source & Target | 87 | 67 | 75 | 60 |

## 4. Conclusion

We have evaluated the effect of EWC on performance after re-training a segmentation model with data from a different domain, and compared it to training on a single dataset and on both domains simultaneously. Adapting to the new domain led to better performance than co-training, and performed slightly better than training on the target domain only. An increase of the EWC weight slightly decreased the performance on the target domain, but in most cases substantially increased performance on the source domain. We can therefore conclude that EWC restricted the degree of catastrophic forgetting but also the ability to adapt to the new domain.

The lower Dice score on the target datasets could be due to the fact that these low-grade glioma are relatively small, and some scans have poor image quality.

More extensive evaluation with different CNN models is needed to know whether and how EWC can enable continuous learning for glioma segmentation. The network size in the first layers may limit the capacity of this network to adapt to slight domain changes, so increasing the encoder size would be a logical next step. Also, besides the EWC weight, the learning rate and the size of the different datasets can be of influence on the degree of catastrophic forgetting. With the results of this study we can at least conclude that pre-training on a related dataset is effective, but in this case it was better to train a specific model for each domain than to combine them.

## Acknowledgments

This work was supported by the Dutch Cancer Society (project number 11026, GLASS-NL), the Dutch Organization for Scientific Research (NWO) and NVIDIA Corporation (by donating a GPU).

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
