# OpenReview forum: "Towards continuous learning for glioma segmentation with elastic weight consolidation"
_MIDL.io/2019/Conference/Abstract — MIDL Abstract 2019_

### Official Review · AnonReviewer1 · 2019-04-30
**Straight-forward application of elastic weight consolidation to brain tumour segmentation**

**Rating:** 3
**Confidence:** 3

**Review:**

This paper contains no methodological novelty but rather a proof-of-concept of the elastic weight consolidation method on two brain tumour segmentation tasks. It can be concluded that EWC works as expected but that if one is interested in the best performing model for each domain, it is better to train two separate networks.

Are the authors aware of the following Medical Imaging meets NIPS Workshop 2018 submission: Baweja et al. 2018, "Towards continual learning in medical imaging"? It also provides a proof-of-concept of EWC on neuroimaging data, albeit not tumour segmenation.

Do the authors have a theory why joint training of source & target seems to perform so much worse on the target domain compared to sequential learning (with or without the EWC penalty)?

---

### Official Review · AnonReviewer2 · 2019-04-30
**Glioma segmentation with a learning strategy that avoids  catastrophic forgetting**

**Rating:** 2
**Confidence:** 3

**Review:**

This study investigates a learning strategy for avoiding catastrophic forgetting in deep networks, with glioma segmentation as application area. Particularly, the authors use elastic weight consolidation to achieve this goal.

The authors fail to cite highly relevant papers ([1,2]), which focus on medical image segmentation (MRI brain to be specific). Indeed, the methodology proposed in those works is the same as in the proposed abstract (i.e., Elastic Weight Consolidation). The authors merely mention that they employ a EWC loss function; they did not give more details. Being the most important part of this work, the authors should have spent more space on this than on the other parts (e.g., the network architecture). Regarding the results, it seems that the choice of the weight of the EWC is very sensitive to the data splitting. While EWC 10 significantly outperform the +Target strategy on the source (split 1), it decreased the performance in the split2. In my opinion, the sole contribution of this work is the application of existing methods (e.g., [2]) to brain tumor segmentation.

[1] McClure P, Zheng CY, Kaczmarzyk J, Rogers-Lee J, Ghosh S, Nielson D, Bandettini PA, Pereira F. Distributed weight consolidation: A brain segmentation case study. In NeurIPS 2018 (pp. 4093-4103).
[2] Baweja C, Glocker B, Kamnitsas K. Towards continual learning in medical imaging. In Medical Imaging meets NIPS workshop, NeurIPS 2018

---

### Decision · Program_Chairs · 2019-05-06
**Acceptance Decision**

Accept